# Micronucleus Assay: The State of Art, and Future Directions

**DOI:** 10.3390/ijms21041534

**Published:** 2020-02-24

**Authors:** Sylwester Sommer, Iwona Buraczewska, Marcin Kruszewski

**Affiliations:** 1Centre for Radiobiology and Biological Dosimetry, Institute of Nuclear Chemistry and Technology, Dorodna 16, 03-195 Warszawa, Poland; i.buraczewska@ichtj.waw.pl (I.B.); m.kruszewski@ichtj.waw.pl (M.K.); 2Department of Molecular Biology and Translational Research, Institute of Rural Health, Jaczewskiego 2, 20-090 Lublin, Poland

**Keywords:** micronucleus assay, genotoxicity, biomonitoring, biological dosimetry, genotoxicity tests, buccal cells, lymphocytes

## Abstract

During almost 40 years of use, the micronucleus assay (MN) has become one of the most popular methods to assess genotoxicity of different chemical and physical factors, including ionizing radiation-induced DNA damage. In this minireview, we focus on the position of MN among the other genotoxicity tests, its usefulness in different applications and visibility by international organizations, such as International Atomic Energy Agency, Organization for Economic Co-operation and Development and International Organization for Standardization. In addition, the mechanism of micronuclei formation is discussed. Finally, foreseen directions of the MN development are pointed, such as automation, buccal cells MN and chromothripsis phenomenon.

## 1. Introduction

Many chemical, physical or biological factors can cause cell death. However, there is a class of factors that not necessarily kill cells, but only damage their genetic material. Such factors are referred to as genotoxins. Although many effective ways to repair DNA damage were discovered, sometimes they fail, which might lead to damage fixation, if the damaged cell survives, and subsequently to its transfer to next generations, if the damaged cell divides.

In a modern world, humans are exposed to different genotoxic agents present in the polluted environment. Hence, tests are needed to determine the level of exposure and health risk. Although many tests classified as “in vivo biomonitoring” are available, a micronucleus test (MN) is one of the best and the most popular [1,2]. The assay is also widely used to test genotoxicity in vitro [1,3,4]. However, it should be emphasized that the concept of MN covers many different techniques that might be used in certain, specific situations [5,6]. In this review, we present various applications of the MN, discuss associated technical challenges and describe future directions of its development.

## 2. Why Do We Need Genotoxicity Testing, Biomonitoring, Cytogenetic Tests and Biological Dosimetry?

Many physical and chemical factors affect stability of our genome and can contribute to the development of civilization diseases, such as cancer, cardiovascular disease, chronic obstructive pulmonary disease or neurodegenerative diseases [4,5,7,8,9]. Genotoxicity testing allows for an assessment of their impact on humans and biota [10,11]. Studies related to genome stability are not an easy task, as they are biased by many confounding factors, often giving inconsistent results or of low statistical strength that enable a direct attribution of the disease to examined factors. Nevertheless, in some cases, such studies could be conclusive, e.g., studies on a role of radon-induced DNA damage in development of lung cancer, impact of PM-10 and PM-2.5 dusts on human health, cytogenetic effects of unhealthy nutrition or effects of risky, unhealthy behaviors and addictions, such as smoking cigarettes or chewing betel quid [12,13,14,15,16].

Nearly 20 types of well-described in vitro or in vivo genotoxicity tests are presently used [10,17]. In vitro assays are used to investigate the potential genotoxic effect of new pharmaceuticals and other medical materials, daily use goods, chemical and physical factors and poisons, etc. [18]. In vivo tests, despite the aforementioned applications, allow for investigating the impact of environmental factors on humans or biota, the impact of the harmful working environment on human health or early genetic changes associated with various diseases development. The most often used in vitro assays include testing of induction of nucleotide mutation, e.g., bacterial reverse mutation assay (Ames test), mammalian cell gene mutation assay or mouse lymphoma assay, and testing of more complex genome changes by cytogenetic methods, such as the sister chromatid exchange assay, analysis of chromosomal aberration frequency, cytokinesis-block micronucleus assay (CBMN) and comet assay (single-cell gel electrophoresis). In addition to the aforementioned applications, in vivo tests allow for investigation of the impact of environmental factors, working environment or changes associated with various diseases. The most important in vivo tests include three cytogenetic methods, namely comet assay, chromosomal aberration assay and different kinds of MN, including CBMN, mammalian erythrocyte MN (EMN) or buccal cells MN (BMN) [17].

Cytogenetic methods are commonly used in vivo for genotoxicity testing (on animals) and in vitro for genotoxicity testing (on cell lines) of compounds of drugs or food chemicals [2,4,19]. Cytogenetic methods are important to search for DNA damage resulting from environmental and occupational exposure to chemical pollutants and physical factors, as ionizing radiation (IR) [2,3]. Although DNA damage induced by IR is processed by different DNA damage repair systems [20], its improper repair may lead to cell death or formation of mutagenic lesions of different complexity. Cytogenetic methods deal with complex type mutations, known as structural mutations (aberrations), such as dicentrics, translocations, acentric fragments, rings and chromatid type aberrations, and a special type of chromosomal damage, known as micronucleus (Mn). Knowing the frequency of changes in genetic material, it is possible to estimate the extent of exposure or even to reconstruct an absorbed dose of ionizing radiation [21,22]. The level of cytogenetic damage to the cells is proportional to the radiation dose and can be read. This procedure is called biological dosimetry and it is usually used to reconstruct the dose in the event of uncontrolled exposure during accidents in medicine or industry [21,22,23]. In this regard, the most commonly used technique is the dicentric chromosome assay due to its high specificity for radiation, but CBMN is also often used, less specific, but easier to perform, faster and easy to automatize [21,23]. Methods used in biological dosimetry are used to indicate exposure to different chemical and physical factors, different than IR. Though biological dosimetry is the main field of CBMN application, the method is also used in theater areas of medicine, e.g., to detect and study genetic and civilization diseases. Several examples of its application in biomedical sciences are given in Table 1.

## 3. How Micronuclei are Formed?

Mn is a small, chromatin containing round-shaped body visible in the cytoplasm of cells [6,30]. Mn is considered to be caused by DNA damage or genomic instability [31]. Mn can occur as a result of natural processes, such as metabolism or aging or can be induced by many environmental factors, hazardous habits and different diseases. The vast majority of factors giving origin to Mn are well recognized and described in literature [6,30,31,32]. The most often mentioned are listed in Table 2.

## 4. Different Types of Micronucleus Assays

Although all types of MN are based on the analysis of micronuclei frequency, they vary in terms of used cells and protocols. The summary is given in Table 3, followed by a more detailed description.

### 4.1. Cytokinesis-Block Micronucleus Assay (CBMN)

The most popular version of MN is the cytokinesis-block micronucleus assay (CBMN) [6,30] (Figure 1). Because Mn is visible only after cell division, the cytochalasin B that inhibits actin filaments polymerization and formation of contractile microfilaments is used to stop cytokinesis [42,43]. However, cytochalasin B does not stop karyokinesis, thus binucleated cells are formed with Mn present in their cytoplasm. The influence of cytochalasin B on cell proliferation and induction of Mn was discussed in the past [6,44]. The conclusion reached indicates that in most cases, the usage of cytochalasin B does not induce additional Mn, hence, the use of cytochalasin B is recommended [5,6,30,45,46,47,48]. This is of special importance when human lymphocytes are used, as their cell cycle may vary among individuals [4]. According to the mathematical model described by Fenech, MN with cytochalasin B applied to block cytokinesis is superior over MN without cytochalasin B because there are less false-negative results when MN using cytokinesis block is used [49].

The CBMN is prevalently performed on human peripheral blood lymphocytes to study in vivo formation of Mn for biomonitoring or biological dosimetry, however, can be performed on different lymphocytes of other species, e.g., rodents, fish, dogs, rabbits, monkey and apes or other cells of different origin [50,51,52]. The CBMN is also very often used on blood samples in vitro to study genotoxic effects of chemicals. [6,53,54,55,56,57]. The information about in vitro genotoxicity testing by MN is gathered, revised and systematized in the Organization for Economic Co-operation and Development (OECD) 487 Guideline [4].

The CBMN provides a comprehensive basis for in vitro investigating of the chromosome damaging potential of chemicals, noteworthy, both aneugenic (changes in the chromosome number in the cell caused by e.g., tobacco smoking, pesticides) and clastogenic changes (structural aberration caused e.g., by ionizing radiation; acridine yellow, benzene, ethylene oxide, arsene, phosphine) can be detected. According to the OECD guideline, cells should be are treated with chemical compounds in three different ways: cultured with cytochalasin B, cultured without cytochalasin B and cultured in the presence of exogenous metabolic activation system, usually prepared from the liver of rodents (S9 fraction). It is impossible to enumerate all applications of the CBMN. The most important applications of CBMN have already been described, but several more are mentioned in Table 4.

CBMN disorders characterization-occupational exposure, although in the basic version of the assay only Mn are scored, the assay can be extended by scoring other biomarkers, as nucleoplasmic bridges, nuclear buds, nuclear blebs, necrotic and/or apoptotic cells [32]. This type of the assay, called CBMN cytome assay, gives additional information about DNA damage and its repair, cytostasis and cytotoxicity [72].

### 4.2. Erythrocyte Micronucleus Assay–the Most Popular In Vivo MN

Erythrocyte micronucleus assay (EMn) was initially performed on immature erythrocytes from bone marrow of young mice and rats [73]. The disadvantage of the assay is that bone marrow examination entails sacrificing rodent life. In addition, potentially confounding factors, such as other nucleated cells (must cells, granulocytes or different types of lymphocytes), are present in the bone marrow [74]. EMn was also performed on cellular material taken from human bone marrow to determine cytogenetic damage after radio- and chemotherapy [75,76,77,78].

Due to the high invasiveness of the method, an alternative approach was developed, based on assessing the frequency of Mn in immature erythrocytes in peripheral blood [1,79]. During maturation, erythrocyte precursor cells lose their nuclei, however, retain Mn formed during the nucleated stage [2]. Immature erythrocytes (also called reticulocytes or polychromatic erythrocytes) can be easily recognized from the mature erythrocytes because they still contain RNA in their cytoplasm [1]. In bone marrow, immature erythrocytes constitute about 50% of all erythrocytes [80,81].

Human erythrocyte precursor cells present in the long bone marrow of control specimens contain nuclei, likely as a result of environmental or occupational exposure or genetic factors (Figure 2). Immature erythrocytes arising from the precursor cells also contain Mn, however, they represent only a few percent of all erythrocytes in peripheral blood [81]. Though frequency of Mn in immature erythrocytes is significantly higher than in mature erythrocytes [1], splenic selection, a process that effectively removes micronucleated erythrocytes from peripheral blood, significantly reduces the frequency of Mn. Splenic selection occurs in rats and humans, also in mice, but to a lesser extent [74,82]. Therefore, in humans, the MN in immature erythrocytes is carried out only on individuals with the spleen removed [13,83]. To increase the assay reliability, the MN in immature erythrocytes in peripheral blood has been automated and is carried out by flow cytometry. With this technique, hundreds of thousands of cells can be analyzed in a reliable time, which allowed overcoming problems with low number of cells available for analysis and Mn splenic selection. Flow cytometry aided scoring of Mn in erythrocytes was validated both in rodents and humans [82,84,85].

### 4.3. Buccal MN (BMm)–Mature but Underused Assay

Although BMm has been used for about 40 years, it seems that only in recent years it gains more interest. The first publications describing this test appeared in the 1980s [86,87]. However, the first publication of the operational protocol in Nature Protocols falls within the last 10 years, after harmonization of the assay by the international HUMNxl group [88,89]. Mn arise in dividing basal cells of oral epithelium but are observed in differentiated cells in the keratinized layer at the buccal surface [90,91]. In addition to Mn, several other cytogenetic biomarkers, including those related to cell death, can be analyzed, which gives more information of the origin of DNA damage, cytostasis and cytotoxicity, somehow analogous to the CBMN cytome assay mentioned earlier [88,91]. This approach was also called the buccal micronucleus cytome (BMCyt) assay [88,91,92].

Mn in the buccal cells are formed in the organism, in rapidly dividing buccal epithelial tissue (Figure 3). Although cells from the oral cavity are exposed to genotoxic or cytotoxic factors by inhalation and food intake to a greater extent than peripheral blood lymphocytes, the background frequency of Mn in buccal cells is very low [93,94]. On the other side, patients undergoing head and neck radiotherapy may serve as a positive control, although some ethical issues must be considered, as in their case the collection of material is more problematic due to lesions and inflammation of the oral mucosa [88,95]. The BMm was used to investigate the impact of nutrition, lifestyle factors (as smoking, drinking alcohol or betel chewing), genotoxin and cytotoxin exposure. Interestingly, correlation was found between the frequency of Mn in buccal cells and increased risk of accelerated aging, certain types of cancer and neurodegenerative diseases [92,96].

BMn seems to be advantageous over other types of MN to study how genotoxic factors affect organisms by inhalation. It is the only method capable to show genotoxic effects of moderate concentrations of radon, as those that are met in unventilated rooms, basements or caves [14,97]. Moreover, the sensitivity of BMn allowed also to show genotoxic effects of work in healthcare, where staff was exposed to very low doses of radiation [98]. BMn is gaining popularity and probably will become a standard cytogenetic test, especially since it is minimally invasive, easy to perform, and cell samples are taken from the oral cavity.

### 4.4. Other Types of MN

Occasionally, the MN is also performed on cells other than lymphocytes, fibroblasts and buccal cells, such as nasal mucosa cells or urine-derived cells [99,100,101]. Both the test objectives and the method of performance remain similar to CBMN or BMn, but these tests have not gained much popularity, so far.

## 5. Visibility by International Organization

The future application of the assay gains more interest since its recognition by international protection, normalization and scientific organizations, such as International Atomic Energy Agency (IAEA) [21] and biological dosimetry network “Running the European Network of Biological and retrospective Physical dosimetry” (RENEB), both considering MN as a fine biodosimetry method [102] or OECD and International Organization for Standardization (ISO), which issued the standards of how to use the MN in genotoxicology and biological dosimetry, respectively [2,4,103]. Finally, the International Human Micronucleus Project (HUMN), which is the biggest database of results of MN in lymphocytes and buccal cells, was established [96].

The IAEA has always considered biological dosimetry, including MN, as an important component of the radiation protection system. Under its auspices, subsequent editions of textbooks on this issue were developed in 1986, 2001 and most recently in 2011 [22,104,105]. Recent publication: “Cytogenetic Dosimeters: Applications in preparedness for and Response to Radiation Emergencies”, in EPR-Biodosimeters counts over 200 pages and contains 350 references, which makes it the most comprehensive compendium of knowledge about biological dosimetry techniques, so far [22]. In addition, the IAEA prepared a series of reports on accidents involving radiation exposure that happened worldwide, with a thorough analysis of what had happened, why and what conclusions could be drawn for the future. In some of these accidents, the CBMN was used to estimate the radiation dose [106].

The IAEA supports the development and sustainability of biological dosimetry methods in member states counties. This is realized by, among others, so-called coordinated research activities (CRAs), which aim at acquisition and dissemination of new knowledge and technology generated through the use of nuclear science, radiation and isotopic techniques [107]. In 2017, the new CRA titled: “Applications of biological dosimetry methods in radiation oncology, nuclear medicine, diagnostic and interventional radiology” (MEDBIODOSE) was launched [108]. The goal of this CRA is to prove the feasibility of biological dosimetry methods in relation to the effects of medical procedures involving ionizing radiation and their improvement. The CBMN is considered as one of the most valuable assays for this purpose and is being used in 7 out of 39 subprojects, covering such topics as, biological dosimetry for individually tailored patients’ treatment, biological dosimetry for genotoxicity assessment, prediction of radiation toxicity of normal tissue by ex vivo irradiation or by in vivo biomarkers, and use of CBMN in cancer risk prediction [29].

The European biodosimetry network RENEB was created as the result of two projects: MULTIBIODOSE and RENEB focused on the implementation of cytogenetic tests in Europe [55,102]. In the frame of the MULTIBIODOSE project, CBMN was harmonized at partner’s laboratories and the automatic and semi-automatic versions of the assay was successfully validated [55]. It was proven that the semi-automatic version of the assay gives results as good as manual scoring. Its continuation, the RENEB project has proved that CBMN can be used in a mass radiation exposure scenario giving results in a reasonable time [109]. Various types of interlaboratory comparisons were carried out, testing individual methods of biological dosimetry and reaction of laboratories to network activation. The capability of biological dosimetry in Europe was also estimated, calculated as the number of samples possible to analyze in the given time [110]. Since RENEB cooperates with biological dosimetry networks in Canada, South America, Japan and Asia, an international sample exchange is possible (and was tested) [111,112], the first step to the global biological dosimetry network.

CBMN is present also in the area of interest of organizations issuing standards on the use of research methods in various aspects of everyday life and industry, such as OECD or ISO [2,4,103]. Their technical documents and/or standards are the basis for method accreditation, theoretically guaranteeing that the laboratories performing tests according to these documents receive objectively correct results, comparable to other accredited laboratories. As far as MN is concerned, the OECD has issued two Technical Guides: “474 OECD Guideline for the Testing of Chemicals, Mammalian Erythrocyte Micronucleus Test” and “OECD Library Test No. 487 OECD Guidelines for the Testing of Chemicals, In Vitro Mammalian Cell Micronucleus Test” [2,4], both dedicated to laboratories examining the genotoxicity of drugs and chemicals in vivo and in vitro. It has to be emphasized that genotoxicity studies are an important part of human clinical trials of new medicaments [82]. In turn, ISO issued the ISO 17099:2014 standard “Radiological protection—Performance criteria for laboratories using the cytokinesis block micronucleus (CBMN) assay in peripheral blood lymphocytes for biological dosimetry” [103].

The HUMN project (www.humn.org) was founded in 1997 to coordinate worldwide research efforts aimed at using MN to study DNA damage in the human population [31,96]. For 15 years, several goals have been achieved: 130 laboratories that published papers on the MN assay were identified and 42 of them were cooperating in HUMN, CBMN protocol and scoring criteria were standardized and the connection between the level of Mn and age, gender and smoking status was established. In addition, the median control level of 6.5 Mn per 1000 binucleated cells was found for CBMN in a database of 7000 donors. The prospective study with the same data indicated that increased Mn frequency is associated with the cancer risk.

In 2007, the HUMN project coordinating group decided to launch a new project considering BMN and the project was called HUMNXL, as Mn are scored in buccal eXfoLiated cells [96]. The questionnaire was sent to 188 laboratories all over the world, based on issued publications using BMN. Fifty-eight laboratories decided to cooperate in the project and a database of more than 5000 subjects was collected. BMN protocol and scoring criteria were established. Factors as age, gender and smoking, which can change the control level of Mn was considered. The control level of Mn in the buccal cells is as low as 1.1/1000 cells (95 % CI 0.70–1.72), but to reduce the variability of Mn mean estimates, it is recommended to score 4000 cells (instead 2000, as it is usually done) [94,96]. The interlaboratory comparison had been done and proved that participants correctly recognized samples from persons who had undergone RT, based on frequency of Mn, nuclear buds and differentiated binucleated cells [95].

## 6. Automation of MN

MN is reliable but a laborious and time-consuming method. One of the ways to speed up the results acquisition is its automation. At the same time, automation increases the reliability of the assay, minimalizing a bias associated with different recognition of Mn by different scorers, different levels of experience and fatigue. Moreover, automation increases the statistical certainty of the results, enabling analysis of a larger number of cells, several thousand or even tens of thousands instead of one, two thousand analyzed in the classic version of the assay [53]. Automation is expected to give higher accuracy and allowed to survey effects on the low observable level, not possible with the manual scoring [82]. However, according to Rodrigues [53], the existing automatic micronucleus scoring systems on microscope slides or by flow cytometry experience technical problems and do not fully meet the requirements of rapid biological dosimetry with a large number of samples to be analyzed.

### 6.1. Automatic/Semiautomatic Scoring by Microscope Aided Systems

Attempts to automate CBMN have been made many times [113,114]. In our subjective opinion, the analysis of microscopic preparations using the image analysis system Metafer (Metasystems, Germany) [115,116] has gained the greatest popularity so far. The system is commercially available (https://metasystems-international.com/). It consists of a motorized microscope, a CCD camera and a computer program that captures pictures of the preparation and searches for double-nucleated cells with or without Mn (Figure 4). The advantage of the program is that it creates a gallery of pictures and allows the researcher to relocate them on a microscope slide to verify the results obtained by the system. The program was used by most participants of the RENEB project, which proved that reconstruction of the radiation dose using automatic or manual counting gave similar results [55]. The program shortens the analysis of preparations by at least 4 times, allowing scoring 2000 cells in 30–40 min instead of 1000 cells in 2 h for manual counting. The software is suitable for Mn analysis for biological dosimetry, but equally well can be used in in vitro tests carried out according to the OECD 487 Guideline.

Microscope aided systems of scoring Mn face several challenges. To perform automated slide-scoring, the cells must be fixed in the mixture of methanol and acetic acid and left for several hours or overnight to dry out [117,118]. This implicates delay in gaining results. Another problem is that to perform a fine automatic analysis of slides, good quality images and the appropriate density of cells are necessary [117,119]. Both too low and too high number of cells increase false Mn positive scores [116,118]. Finally, only the nuclei are stained on the preparation, not cytoplasm [53]. This leads to the situation that only possible recognition of mononucleated, binucleated and polynucleated cells is achieved by mathematical algorithms.

### 6.2. Flow Cytometry and Imaging Flow Cytometry Aided Mn Scoring

Besides the microscopic analysis of binucleated cells, attempts had been made to use a flow cytometry method to analyze the frequency of Mn in lymphocytes or other nucleated cells [120,121]. However, the method faces two major obstacles. First, the cells must be lysed because micronuclei are only recognized base on their size. Objects smaller than cell nuclei are gated, but it is uncertain whether these are not debris or apoptotic bodies or necrosis residues [120,122]. Secondly, cytochalasin B is not used in this kind of test because the cells are lysed and thus binucleated cells cannot be analyzed. This approach might underestimate the number of Mn [123].

Problems encountered when using automated slide-scoring systems or flow cytometry are solved by the imaging flow cytometry (IFC) method that is claimed to be a combination of flow cytometry with automatic cell image analysis [53]. The system analyzes the cells flowing individually through the lasers beam, and in the meantime, every object in the solution is photographed. Based on the pictures, objects as mononucleated, binucleated, polynucleated cells, Mn, nuclear bridges and nuclear buds, as well as apoptotic bodies can be recognized. The high throughput is one of the system features [53].

### 6.3. Automatization of EMn

EMn has been automatized by using the automated analysis of images of microscopic preparations or flow cytometry [82,124,125]. The analysis of microscopic slides is described above, thus here we will briefly describe the use of a flow cytometry technique. The method allows analysis of both columns fractionated bone marrow and peripheral blood samples [126,127]. Cells are stained simultaneously with propidium iodide/Hoechst and anti-CD71 antibody (transferrin receptor characteristic for immature erythrocytes) and anti-platelet-specific antibody (anti-CD61), both differentially labeled with fluorochromes. Single, unaggregated cells are gated, excluding those exhibiting anti-platelet associated fluorescence. Then, CD-71 positive cells are recognized and Mn stained with propidium iodide/Hoechst are scored. The method has been transformed into a commercially available kit format (e.g., In vitro MicroFlow, Litron Laboratories, Rochester, NY, USA). The assay has been widely used [128,129]. It has been investigated and validated for different species samples (mice, rats, human) by The Collaborative Study Group of the Japanese Environmental Mutagen Society [130,131]. The assay is important for human population studies because performing it in blood samples makes it not invasive when compared with bone marrow sampling. Automation made the blood erythrocytes examination feasible and it is possible to consider the damage found in human blood, as an index of damage in the bone marrow.

## 7. Chromothripsis

Chromothripsis phenomenon—massive changes in the genetic material—hundreds of rearrangements, confined to single or a few chromosomes are important in cancer and congenital diseases development and perhaps in rapid karyotype evolution [39,132,133,134]. A chromothripsis has been found in 2–3% of cancers of all types including 25% of bone cancers and 18% of late-stage neuroblastoma [39,133,134,135,136,137,138,139]. It is suggested that chromothripsis may be the result of one catastrophic event of massive breakage closely followed by rearrangements and repair processes [140,141]. The mechanisms involved and the meaning of phenomena are not fully understood, but the most accepted model of how chromothripsis can be confined to the single chromosome of the chromosome arm is the Mn model [135,142].

Mn has been considered as a result of induction of different kind of DNA damage and a good indicator of chromosomal instability which is always present in pre-cancer cells [30,31,32,33,72]. The replication machinery and DNA repair are impaired in Mn, which in a short time leads to extensive DNA damage [31,142]. Then, if the Mn is included in the cell nucleus again there is a lot of rearranged chromosome material integrated, in most cases derived from a single chromosome and chromothripsis can be initiated. The traditional view of Mn faith is that cells bearing Mn are doomed to death in a close future, after few division latest, as a result of chromosome instability [123], or Mn themselves can be extruded out of the cells [35]. Novel experiments utilizing long-term live-cell imaging show that Mn can be incorporated to the daughter cells during mitosis and stably remains in cytoplasm [142,143,144]. Then, Mn material sometimes is reincorporated to the arising nucleus, which was revealed e.g., by photoactivation of micronuclear chromatin [30,142]. Chromothripsis phenomenon changes traditional thinking about Mn as a passive depiction of genetic material damage processes to the active players in formation of DNA lesions [31].

## 8. Conclusions

There are obvious reasons to think that Mn is important for future use as one of the most reliable, well established and feasible genotoxicity tests [2,3,4,96]. BMn (described in more detail in Chapter 4) will gain more importance, as a convenient and reliable biomonitoring test, which in the most exposure scenario may be more sensitive than the standard CBMN. The MN is easy to automate, works to fully automatize the method, has been carried out for some time on at least 3 major platforms: automated slide-scoring, flow cytometry and imaging flow cytometry. Certainly, further attempts will be made to determine the relationship between the phenomenon of Mn formation and the development of cancer.

## Figures and Tables

**Figure 1 ijms-21-01534-f001:**
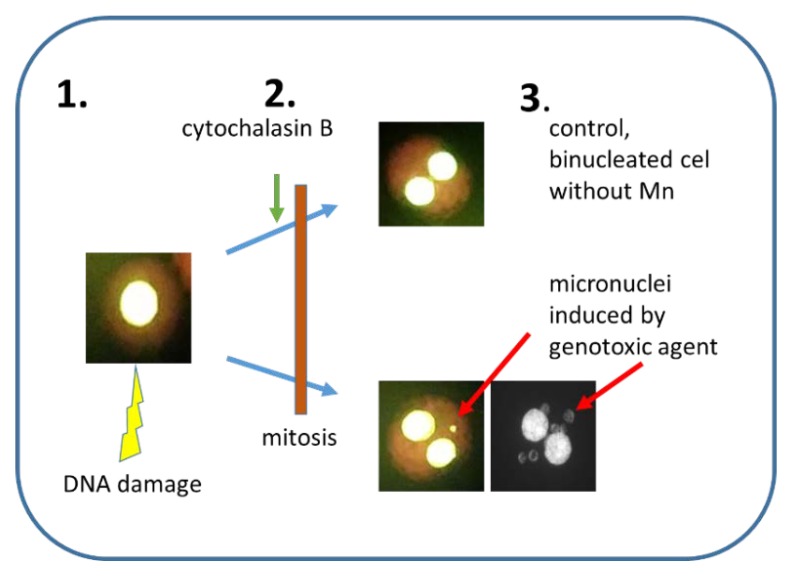
A principle of cytokinesis-block micronucleus assay. 1. Nucleus with damaged DNA. 2. Inhibition of cytokinesis by the addition of cytochalasin B. 3. The Mn frequency is scored in binucleated cells only. Upper part—control binucleated cells without Mn, lower part—two binucleated cells with 1 or 6 Mn visible in the cytoplasm.

**Figure 2 ijms-21-01534-f002:**
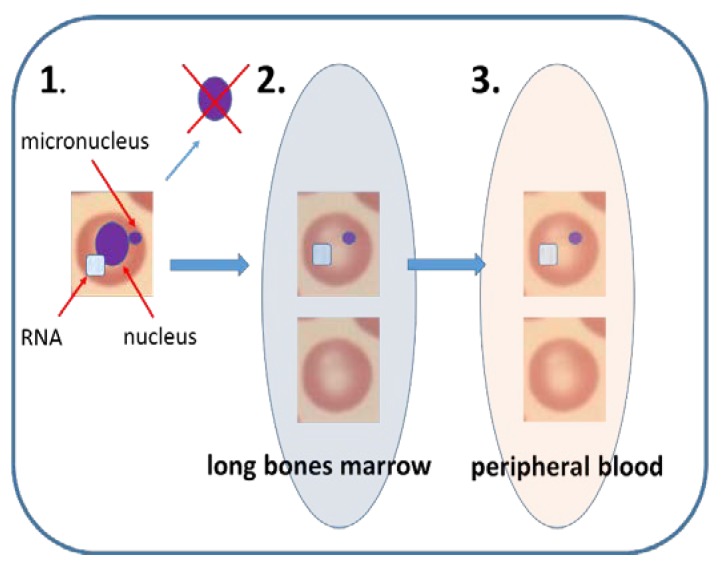
Mammalian erythrocyte micronucleus assay. (1). Immature erythrocyte in bone marrow contains nucleus and RNA in its cytoplasm. When DNA damage is induced in vivo, micronucleus can arise in the nucleated erythrocyte. When nucleus is excluded during erythrocyte maturation the micronucleus stays in the cytoplasm. (2). In bone marrow, immature erythrocytes consist of around 50% of all erythrocytes. Occasionally these immature erythrocytes may contain Mn. (3). Sometimes the immature erythrocytes are released to peripheral blood, where they constitute less than 5% of all erythrocytes. The immature erythrocytes in blood can be recognized due to their specific surface receptors or RNA content. Flow cytometry technique makes the EMn feasible in peripheral blood of rodents and humans.

**Figure 3 ijms-21-01534-f003:**
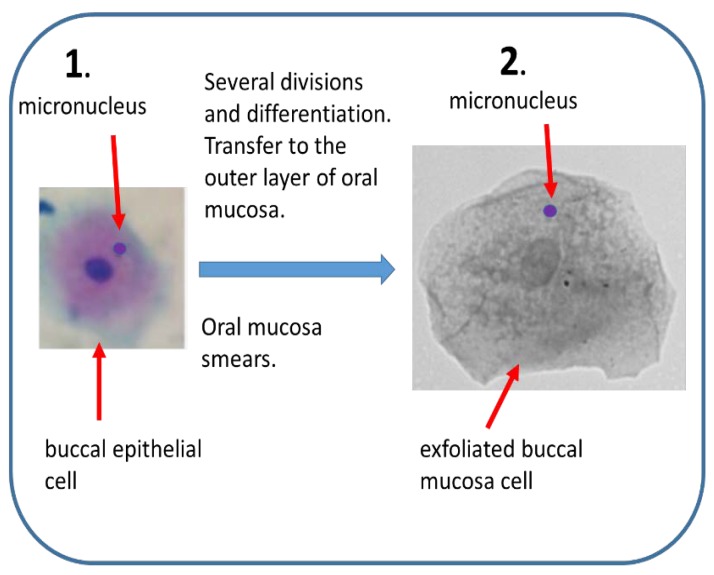
Buccal micronucleus assay. (1) Mn are induced in vivo by genotoxic agents in rapidly dividing buccal epithelial tissue. Epithelial cells differentiate and move towards outer layer of oral mucosa. (2) Mn frequency can be estimated in smears of exfoliated buccal cells.

**Figure 4 ijms-21-01534-f004:**
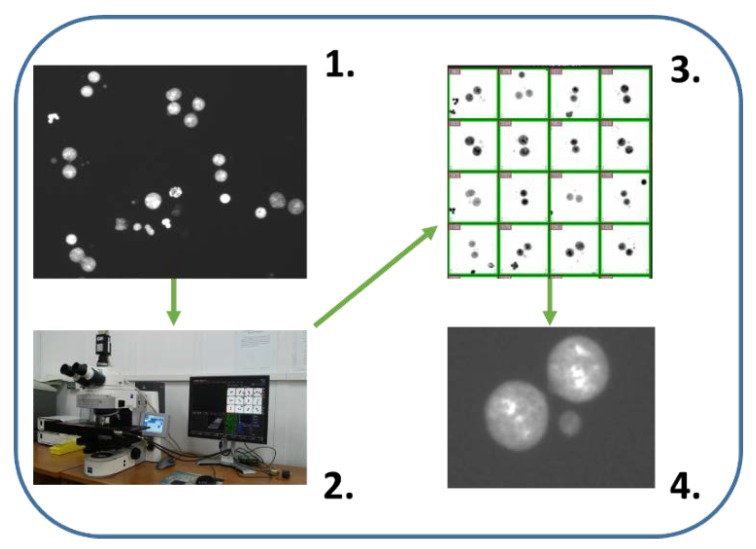
Automatic/semiautomatic scoring by microscope aided systems (Metasystems). (1) Nuclei of cytokinesis-block cells are visible. System recognizes binucleated cells when both nuclei are the same size and they are in proximity to each other. (2) Motorized microscope with image analysis system safe galleries of binucleated cells and recognize Mn in an automatic manner. (3) Gallery of binucleated cells. Mathematical algorithms to recognize binucleated cells implemented in Metasystems has been used many times with good results [55,115,116]. (4) Each cell from the gallery can be localized and e.g., false positive Mn can be verified in high magnification. Such semiautomatic scoring gives better results than automatic scoring.

**Table 1 ijms-21-01534-t001:** Examples of the applications of cytokinesis-block micronucleus assay to detect and study genetic and civilization diseases.

Diagnose genetic disorders like Fanconi anemia, ataxia telangiectasia or various autoimmune diseases [24,25,26].
Evaluation of individual susceptibility to the effects of exogenous or endogenous genotoxic agents [3,12].
Assessment of the risk of developing cancer and other chronic diseases [12].
Prediction assay for a radiation side effect (normal tissue reaction) of patients with different cancers subjected to RT and post RT follow up [27,28,29].
Quantification of in vitro genotoxicity of different schemes of RT [29].

**Table 2 ijms-21-01534-t002:** Possible origin of micronuclei.

**Acentric chromosome fragments**
Acentric chromosome fragments result from unrepaired DNA strand breaks or misrepaired DNA strand breaks, DNA strand cross-links or adducts leading to chromosome or chromatid type aberrations, e.g., polycentric chromosomes usually accompanied with acentric fragments; [22,31,32,33].
**Malsegregation of chromosomes**
A whole chromosome lagging behind during mitosis or numerical chromosome aberrations, as a result of centromere dysfunction; kinetochore dysfunction, spindle dysfunction. Aging of women, when some chromosomes X are excluded from the nucleus [22,31,32,33].
**Dicentric chromosome breakage**
Polycentric chromosomes, when spread between opposite cells, may break in many pieces, giving rise to Mn or/and broken unprotected chromatid ends [31,32,33,34]. Unprotected chromatid ends are susceptible to different reorganization processes and can start breakage–fusion–bridge cycles [34], that give rise to Mn formation and are considered as one of the mechanisms of chromosome instability of cancer cells [34].
**Chromosome instability**
Chromosome instability is a condition in which cells gain changes in their genome at a high rate. Chromosome instability is often displayed by pre-neoplastic and cancerous cells, which usually show a high frequency of Mn [31,35]. It is accepted that Mn is a good indicator of chromosome instability [31].
**Aggregation of double minutes (DB)**
DB are small acentric and telomere-free extrachromosomal bodies composed of circular DNA [31,36,37]. They have been observed in many kinds of tumors including breast, lung, ovary, colon and neuroblastoma [36,37,38]. DB are the manifestation of genomic instability and recently they have been linked to chromothripsis phenomenon [39]. They carry multiple copies of amplified genes, usually oncogenes or genes involved in drug resistance. Many copies of DB can be found in a single cell often stuck to the chromosomes [40]. DB, when detached from chromosomes, can aggregate and form Mn [41].

**Table 3 ijms-21-01534-t003:** Types of micronucleus assays.

Type of Test	Cells Used	Purpose	Short Characteristics
**Cytokinesis-block micro-nucleus assay (CBMN)**	human, rodents, rabbit, fish, dogs, primates, etc. lymphocytes or cell lines.	biomonitoring; biological dosimetry; in vitro or in vivo genotoxicity; biological experiments where cytogenetic damage is assessed.	Mn is scored in binucleated cells, where cytokinesis is stopped by addition of cytochalasin B. The most popular in vitro and in vivo MN.
**Mammalian Erythrocyte MN**	human, rodents, rabbit, fish, dogs, primates, immature erythrocytes.	in vivo genotoxicity of chemicals, drugs or harmful condition; biomonitoring.	Test performed usually on young rodents, but biomonitoring of the human population based on peripheral blood is possible. When performed in peripheral blood erythrocytes splenic selection must be considered.
**Buccal MN**	human epithelial buccal cells.	impact of nutrition; lifestyle habits, such as smoking and drinking alcohol; genotoxic exposure; cytotoxic exposure; risk of accelerated aging, certain types of cancer and neurodegenerative diseases.	Incoming, little invasive in vivo test. Suitable for biomonitoring.
**MN in other cell types**	nasal mucosa cells, urine-derived cells.	biomonitoring; genotoxicity; prognosis of certain cancers.	Not very popular, although there are new publications.

**Table 4 ijms-21-01534-t004:** Application of CBMN.

basic research on DNA damage and repair [58,59];
radiosensitivity studies of various groups, whether healthy or with genetic disorders [24,60,61,62];
attempts to link the radiosensitivity with the radiation reaction of normal tissues in persons undergoing radiotherapy [63,64];
predictive tests of neoplastic disease [60,65,66];
characterizations of cytogenetic damage during chemo- and radiotherapy [67,68];
biomonitoring of the environment or occupational exposure [32,69,70,71].

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
