# Peer review of "Micronucleus Assay: The State of Art, and Future Directions"

_ijms, 2020, doi:10.3390/ijms21041534_

Round 1

Reviewer 1 Report

The paper was improved and I have only one minor comment:

- Page 2, line 51: "or alterations associated with early genetic changes associated with various diseases development" should be "or early genetic changes associated with various diseases development".

Author Response

Dear reviewer

Thank You very much for Your acceptance of the manuscript. The sentence from line 51 has been changed according Your advice.

Reviewer 2 Report

Thank you for considering the remarks in the revised version of your manuscript. All of the comments have been satisfactorily answered and mostly also implemented in the current version.

Author Response

Dear reviewer

Thank You very much for Your acceptance of the manuscript.

Reviewer 3 Report

All comments are taken into account by the authors. I believe that in the presented form the manuscript can be published.

Author Response

(The authors gave the same response as above.)
